# Continuous Diffusion Models with Explicit Score Matching for Highly Efficient Anomaly Detection

## Abstract

Diffusion models have proven to be highly effective in generating high-quality reconstructed images, making them ideal for the rigorous requirements of reconstruction-based anomaly detection systems. While continuous diffusion models unify discrete implementations, existing methods predominantly rely on denoising score matching (DSM), as directly acquiring explicit scores remains challenging. In this study, we introduce a novel diffusion framework utilizing explicit score matching (ESM) via a dual-stream neural network, trained by maximum likelihood estimation. Based on the first systematic comparison between DSM and ESM paradigms, variance-guided diffusion process is developed to further improve the performance. Comprehensive experimental evaluations confirm the superior anomaly detection capabilities and computational efficiency of the proposed system. The framework's flexibility allows seamless integration with existing diffusion models, offering a potential pathway for broader applications in generative tasks.

## 1 Introduction

Unsupervised learning-based methods, trained solely on nondefective samples, are more effective in recognizing unknown industrial defects to meet the high precision requirement, taking into special account that defective samples usually are rare, unpredictable and have unclear defective patterns Cao et al. (2024); Liu et al. (2023). Existing unsupervised anomaly detection methods can be categorized into three main types: sample synthesis methods, which generate artificial defective samples from normal training data for training anomaly detectors; discriminative feature embedding strategies identify anomalies by comparing discriminative feature representations; and image reconstruction paradigms are highly suitable for industrial applications as they achieve high pixel anomaly-level localization accuracy by reconstructing defective inputs into normal ones and analyzing reconstruction errors. Before the appearance of diffusion model, most of these approaches typically employ encoder-decoder architectures such as auto-encoders (AE)Rumelhart et al. (1986), variational auto-encoders (VAE)Kingma (2013), and generative adversarial networks (GAN)Goodfellow et al. (2020) for reconstruction. In comparison, recent advancements in generative modeling have shown that diffusion models outperform traditional methods in many computer vision applications, leading in both image generation fidelity and anomaly detection accuracy.

Currently, diffusion models are mainly divided into two types: discrete diffusion models, such as denoising diffusion probabilistic modelsHo et al. (2020) (DDPM), and continuous diffusion models, namely score-based modelsSong et al. (2021b). In practice, discrete diffusion models are predominantly used and achieve the best performance, while continuous diffusion models are less explored. Actually, discrete diffusion models can be derived from continuous diffusion models. For example, DDPM Ho et al. (2020) can be considered a special form of score-based model Song et al. (2021b). However, nearly all existing diffusion models employ denoising score matching (DSM) to approximate the explicit score matching (ESM), which will no doubt result in performance degradation. This phenomenon originates from two aspects: there is no actual score for neural network training in present diffusion framework and ESM could not be calculated in the inference procedure for nearly all existing diffusion models.

To address this issue, we propose a new framework for diffusion models that simultaneously predicts the mean and variance via a dual-stream network, enabling the estimation of the actual score (by assuming the feature vectors following Gaussian distribution with diagonal covariance, refer to UAEMao et al. (2020)). This framework solves the problem mentioned above and allows one to obtain the true score. Unlike nearly all existing methods used in score-based models, we develop a new training approach based on maximum likelihood estimation. The performance of our proposed methods is verified using open datasets. Our contributions can be summarized as follows:

1. We propose an ESM-based diffusion framework for anomaly detection, with corresponding loss function based on maximum likelihood estimation to train a dual-stream network, which learns to predict the mean and variance to obtain the actual score for the continuous diffusion model.

2. As our methods can estimate the actual score, we investigate the difference between ESM and DSM through defect reconstruction for the first time, to the best of our knowledge. The results turn out that the impact of approximation of ESM by DSM is not neglected.

3. We propose a new diffusion method that leverages variance in the diffusion process to achive SOTA performance with low computation cost, leading to the emergence of new diffusion methods.

4. Experiments on the MVTec-AD Bergmann et al. (2019), VisA Zou et al. (2022), and MPDD Jezek et al. (2021) datasets confirm the effectiveness and efficiency of our proposed methods.

## 2 RELATED WORK

Diffusion models have made significant advances in the field of image generation, demonstrating exceptional performance in various downstream tasks. SGM Song et al. (2021b) was proposed, which unifies diffusion models using stochastic differential equations to represent diffusion and denoising processes. However, SGM is trained by conditional denoising score matching(DSM) as the actual score(ESM) could not be obtained. For simplicity, we use DSM to denote denoising score matching and ESM to denote explicit score matching.

The diffusion model is composed of the diffusion process(i.e., the noising process) and the denoising process. Taking the Score-Based Generative Model (SGM) Song et al. (2021b) as an example, it introduces the diffusion process of stochastic differential equations (SDE) is as follows:

$$\mathrm{d}x = \underbrace{-\frac{\beta_t}{2}x\,\mathrm{d}t}_{=f(x,t)} + \underbrace{\sqrt{\beta_t}\,\mathrm{d}\mathbf{w}}_{=g(t)}, \tag{1}$$

where $t \in [0,1]$, $\mathrm{d}\mathbf{w}$ is a standard Wiener process, and the drift coefficient $f(x,t)$ and the diffusion coefficient $g(t)$ are determined according to the chosen diffusion process. Through this process, noise is gradually added to the original data, transforming it into Gaussian noise as $t \to 1$.

The corresponding reverse process (i.e., denoising process) can be written as:

$$\mathrm{d}\tilde{x} = -\beta_t \left[ \frac{\tilde{x}}{2} + \frac{\partial \log\left(p_t(\tilde{x})\right)}{\partial \tilde{x}} \right] \mathrm{d}t + \sqrt{\beta_t}\mathrm{d}\bar{\mathbf{w}}, \tag{2}$$

where $\mathrm{d}\bar{\mathbf{w}}$ is also a standard Wiener process and $\tilde{x}$ represents the feature vector in denoising process. $\frac{\partial \log(p_t(\tilde{x}))}{\partial \tilde{x}}$ is the score of the marginal distribution $p_t(\tilde{x})$, and $\beta(t) \in (0,1)$. Neural network is trained by score estimation, which necessitates the use of score matching methods. Score matching can be categorized into two main variants: explicit score matching (ESM) and denoising score matching (DSM). The formula of score matching in the denoising process eq. (2) is known as explicit score matching (ESM):

$$\frac{\partial \log\left(p_t(\tilde{x})\right)}{\partial \tilde{x}} = -\frac{\tilde{x} - \mu_t}{\sigma_t^2}. \tag{3}$$

where $\mu_t$ and $\sigma_t^2$ are the mean and variance of $\tilde{x}$, here $\tilde{x}$ is assumed following Gaussian distribution, i.e. $\tilde{x} \sim N(\mu_t, \sigma_t^2)$, as the same as that of UAEMao et al. (2020). even under Gaussian assumption, DSMcould not estimate the ESM, as variance $\sigma_t^2$ is unknown. SGM Song et al. (2021b) uses denoising score matching (DSM) with conditional probabilities instead:

$$\frac{\partial \log(p_t(\tilde{x} \mid x))}{\partial \tilde{x}} = -\frac{\tilde{x} - \mu_t}{1 - \bar{\alpha}_t}, \tag{4}$$

where $\bar{\alpha}_t = e^{-2 \int_0^t \beta(s)ds}$. Thus, the difference between ESM and DSM is the denominator of eq. (3) and eq. (4), along with difference between the loss functions of ESM (eq. (10)) and DSM (refer to Song et al. (2021b)). Then, corresponding to the diffusion process (eq. (1)), the denoising process formula based on ESM is as follows:

$$\mathbf{d}\tilde{x} = -\beta(t) \left( \frac{\tilde{x}}{2} - \frac{\tilde{x} - \mu_t}{\sigma_t^2} \right) \mathbf{d}t, \tag{5}$$

where the random noise $\bar{\mathbf{w}}$ is omitted and the stochastic differential equation(SDE) is simplified into ordinary differential equation(ODE).

## 3 METHOD

### 3.1 DIFFERENCE ANALYSIS

we define the $l$-difference as $DSM_l = ||E_{p_t(\tilde{x},x)}[(s_\theta(\tilde{x}) - \frac{\partial log(p_t(\tilde{x}|x))}{\partial \tilde{x}})^l]||_1$ and $ESM_l = ||E_{p_t(\tilde{x})}[(s_\theta(\tilde{x}) - \frac{\partial log(p_t(\tilde{x}))}{\partial \tilde{x}})^l]||_1$, here the exponential $l$ means element-wise exponential.

then

$$
\begin{aligned}
DSM_l &= ||E_{p_t(\tilde{x},x)}[(s_\theta(\tilde{x}) - \frac{\partial log(p_t(\tilde{x}|x))}{\partial \tilde{x}})^l]||_1 \\
&= ||E_{p_t(\tilde{x},x)}[\sum_k C_l^k(-1)^k s_\theta(\tilde{x})^{l-k} \cdot (\frac{\partial log(p_t(\tilde{x}|x))}{\partial \tilde{x}_i})^k]||_1 \\
&= ||\sum_k C_l^k(-1)^k \int_{\tilde{x}} \int_x p_t(\tilde{x},x) s_\theta(\tilde{x})^{l-k} \cdot (\frac{\partial log(p_t(\tilde{x}|x))}{\partial \tilde{x}_i})^k dx d\tilde{x}||_1 \\
&= ||\sum_k C_l^k(-1)^k \int_{\tilde{x}} s_\theta(\tilde{x})^{l-k} \cdot \int_x \frac{1}{p_t(\tilde{x}|x)^{k-1}} (\frac{\partial p_t(\tilde{x}|x)}{\partial \tilde{x}_i})^k p(x) dx d\tilde{x}||_1
\end{aligned}
\tag{6}
$$

In comparison,

$$
\begin{aligned}
ESM_l &= ||E_{p_t(\tilde{x})}[(s_\theta(\tilde{x}) - \frac{\partial log(p_t(\tilde{x}))}{\partial \tilde{x}})^l]||_1 \\
&= ||E_{p_t(\tilde{x})}[\sum_k C_l^k(-1)^k s_\theta(\tilde{x})^{l-k} \cdot (\frac{\partial log(p_t(\tilde{x}))}{\partial \tilde{x}_i})^k]||_1 \\
&= ||\sum_k C_l^k(-1)^k \int_{\tilde{x}} p_t(\tilde{x}) s_\theta(\tilde{x})^{l-k} \cdot (\frac{\partial p_t(\tilde{x})}{\partial \tilde{x}_i})^k \frac{1}{p_t(\tilde{x})^k} d\tilde{x}||_1 \\
&= ||\sum_k C_l^k(-1)^k \int_{\tilde{x}} s_\theta(\tilde{x})^{l-k} \cdot \frac{1}{p_t(\tilde{x})^{k-1}} (\int_x \frac{\partial p_t(\tilde{x}|x)}{\partial \tilde{x}} p(x) dx)^k d\tilde{x}||_1.
\end{aligned}
\tag{7}
$$

It is obvious that only when $k = 1$ and $k = 0$, the corresponding components in the above equations are equal for arbitrary $p_t(\tilde{x}|x)$ and $p_t(\tilde{x})$ neglecting some functions not related to the neural network $\theta$, and $l = 2$ is just the case in Vincent (2011). For $l > 2$, $ESM_l$ usually is not equal to $DSM_l$ for the arbitrary distribution $p_t(\tilde{x})$ and $p_t(\tilde{x}|x)$.

### 3.2 ESM BASED CONTINUOUS DIFFUSION MODEL

In this work, we propose the framework for the detection of anomalies with explicit score matching (ESM) to improve performance, illustrated in fig. 1. This framework could be applied to nearly all

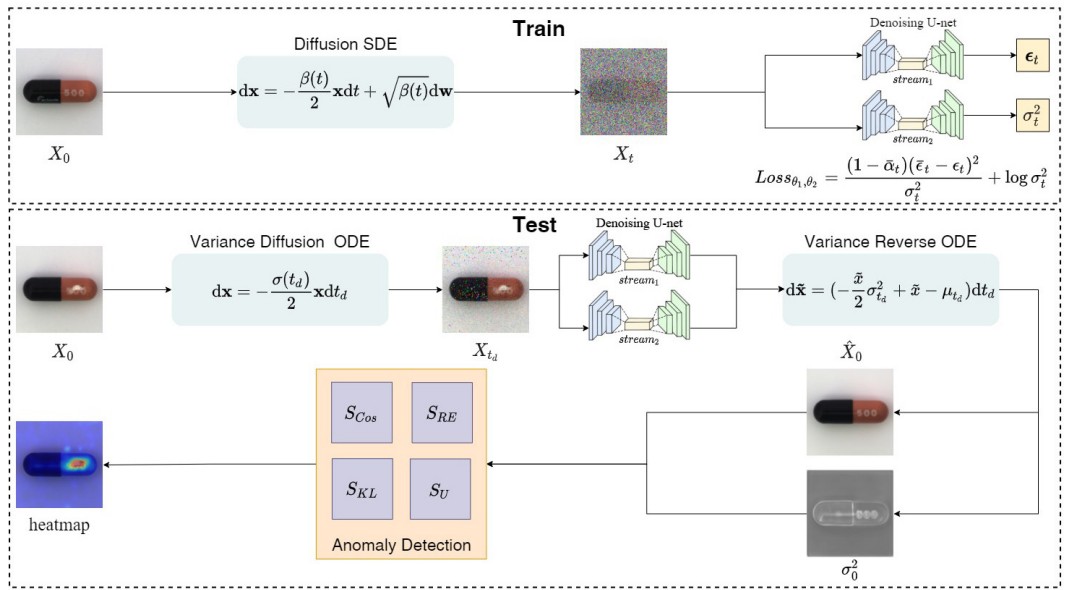

Figure 1: The flowchart of the proposed framework. $X_0$ and $X_t$ represent the input image and the noisy image, respectively. $\epsilon_t$ is the noise expectation, and $\sigma_t^2$ is the variance. $\hat{X}_0$ denotes the reconstructed image, and $\sigma_0^2$ is the variance image at time step $t = 0$. $S_{Cos}$, $S_{KL}$, $S_{RE}$ and $S_U$ stands for the four kinds of anomaly score generated by four detectors. The details are discussed in section 3.

existing diffusion models, as follows: copy the structure of the existing neural network and combine it with the original network to form a dual-stream network, where these two independent streams predict $\mu_t$ and $\sigma_t$ of the image $X_t$ respectively, where $X_t$ is assumed following Gaussian distribution with diagonal covariance.

In training, Gaussian noise is added to defect-free images, which are then processed by the dual-stream denoising network to predict both the mean and variance, thereby modeling the true data distribution (see the upper part of fig. 1 ). In testing, the input image $X_0$ is fed into the trained dual-stream denoising network, where diffusion process could use either a traditionally defined variance $\beta_t$ or our extended approach with estimated variance $\sigma_t$ as the diffusion parameter (see the middle part of fig. 1 ). Finally, four detectors are employed for defect localization, among which a newly proposed detection head explicitly leverages variance information for enhanced accuracy (the bottom part of fig. 1).

### 3.3 LOSS FUNCTION

Assuming that the feature vector $X_t$ following Gaussian distribution with diagonal covariance at time $t$, the dual-stream network in fig. 1 is utilized to predict the expectation $\mu_{\theta_1}(X_t, t)$ and variance $\sigma_{\theta_2}^2(X_t, t)$ separately, where $\theta_1$ and $\theta_2$ are the parameters of stream 1 and stream 2, respectively. In the text, we simplify $\mu_{\theta_1}(X_t, t)$ and $\sigma_{\theta_2}^2(X_t, t)$ as $\mu_t$ and $\sigma_t^2$ without confusion, $t = 0, \cdots, 1$. Here the dual-stream network structure is chosen as $\mu_t$ and $\sigma_t^2$ are two independent parameters for Gaussian distribution. Otherwise, additional loss should be applied to ensure this independence.

As there are no labels corresponding to $\sigma_t$ for training, the neural network could learn to predict $\mu_t$ and $\sigma_t^2$ through maximum likelihood estimation(UAE Mao et al. (2020)), as follows:

$$\theta_1, \theta_2 = argmax_{\theta_1, \theta_2} p_t(X_t). \tag{8}$$

eq. (8) means that even for the noised image $X_t$, its probability should be higher than other feature vectors, e.g. random noise. By taking the minus logarithm of $p_t(X_t)$, the loss function $Loss_{\theta_1, \theta_2}$ is

as follows:

$$Loss_{\theta_1,\theta_2} = -2\log p_t(X_t)$$
$$= \frac{(X_t - \mu_t)^2}{\sigma_t^2} + \log \sigma_t^2 + C, \tag{9}$$

where $C$ is only a function of $X_t$ and can be omitted in training.

For computational implementation, the continuous diffusion process is typically discretized. Following the transformation in DDPM, we have the discrete form $X_t = \sqrt{\bar{\alpha}_t}X_0 + \sqrt{1-\bar{\alpha}_t}\bar{\epsilon}_t$ and $\mu_t = \sqrt{\bar{\alpha}_t}X_0 + \sqrt{1-\bar{\alpha}_t}\epsilon_t$, where $\alpha_t = 1 - \beta_t\triangle t$, $\bar{\alpha}_t = \prod_{s=1}^T \alpha_{\frac{s}{T}}$ and $\bar{\epsilon}_t \sim \mathcal{N}(0, \mathbf{I})$. $\epsilon_t$ is the noise expectation estimated by neural network. Then, the loss function $L_{\theta_1,\theta_2}$ could be simplified as follows:

$$Loss_{\theta_1,\theta_2} = \frac{(1-\bar{\alpha}_t)(\bar{\epsilon}_t - \epsilon_t)^2}{\sigma_t^2} + \log \sigma_t^2. \tag{10}$$

Compared with existing diffusion models (based on DSM), $Loss_{\theta_1,\theta_2}$ includes a second-order term $\sigma_t^2$, and it could be considered as the general form of the loss functions of existing diffusion models.

In test, $\mu_t$(obtained from $\epsilon_t$) and $\sigma_t^2$ are substituted into eq. (3) to calculate ESM.

### 3.4 VARIANCE-GUIDED DIFFUSION METHOD

Nearly all existing diffusion models use the manual $\beta_t$ for the diffusion and denoising process without considering the characteristics of input image. Building on the ESM framework, we propose a variance-guided diffusion process that dynamically adjusts the diffusion trajectory. Using the proposed method as above, the estimated variance $\sigma_t^2$ could be incorporated into the diffusion process and denoising process to replace $\beta_t$, denoted as variance-guided diffusion process. The variance-guided diffusion process is as follows:

$$\mathbf{d}x = -\frac{\sigma_t^2}{2}x\mathbf{d}t + \sigma_t\mathbf{d}\mathbf{w}. \tag{11}$$

Here, $\beta(t) = \sigma_t^2$ could be seen as the solution of the following differential function,

$$2\sigma_t d\sigma_t = -\sigma_t^4 dt + \sigma_t^2 dt. \tag{12}$$

This function comes from the truth that the variance $\sigma_t^2$ at time $t$ could be estimated from the initial variance $\sigma_0^2$ of the input and independent noise, under Gaussian assumption. The detail of the function could be found in Appendix. Thus, the $\beta(t)$ is not stochastic and SDE holds for eq. 11. Then, the corresponding denoising process is

$$\mathrm{d}\tilde{x} = -\sigma_t^2 \left[ \frac{\tilde{x}}{2} + \frac{\partial \log(p_t(\tilde{x}))}{\partial \tilde{x}} \right] \mathrm{d}t$$
$$= (-\frac{\tilde{x}}{2}\sigma_t^2 + \tilde{x} - \boldsymbol{\mu}_t)\mathbf{d}t \tag{13}$$

In this way, the variance $\sigma_t^2$ appears in the numerator instead of the denominator in eq. (13), making the denoising process more stable, especially when $\sigma_t^2$ is small. Given the pseudo-random nature of the stochastic noise and the estimation error in the initial variance, we employ the network-estimated variance in the actual program rather than directly using the solution of eq.12 as $\beta(t)$.

### 3.5 ANOMALY DETECTION AND LOCALIZATION

To detect and localize the defects, four detectors are applied, and the anomaly score $\mathcal{S}$ is as follows:

$$\mathcal{S} = u\mathcal{S}_{Cos}(X_0, \hat{X}_0) + v\mathcal{S}_U + w\mathcal{S}_{KL} + r\mathcal{S}_{RE}. \tag{14}$$

Here, $\mathcal{S}_{cos}$ is the cosine similarity for multi-layer features $F$ (obtained by the pre-trained DINO Zhang et al. (2022) model) between the input image $X_0$ and reconstructed image $\hat{X}_0$. $\mathcal{S}_U = \frac{(X_0 - \hat{X}_0)^2}{\sigma_0^2}$ following the idea of UAE Mao et al. (2020). $\mathcal{S}_{KL}$ is the Kullback-Leibler (KL) divergence between $X_0$ and $\hat{X}_0$, with $\mathcal{S}_{KL} = KL(p(X_0)||p(\hat{X}_0))$ being the spatial restoration score

(following the methodology presented in Shin et al. (2023a)). Note that $\mathcal{S}_U$ and $\mathcal{S}_{KL}$ could only be applied by our proposed methods, which could provide $\sigma_t^2$.

Finally, the final anomaly score $S_a$ is obtained to determine the existence of defects,

$$S_a = \frac{1}{N_a} \sum_{i \in Top_{N_a}} S_i, \tag{15}$$

where $S_i$ is the $i$th pixel value of $\mathcal{S}$, and $Top_{N_a}$ is the collection of the top $N_a = 250$ maximum pixel values of $\mathcal{S}$.

## 4 EXPERIMENT

The experiments are conducted on three popular public datasets, MVTec-AD Bergmann et al. (2019), VisA Zou et al. (2022), and MPDD Jezek et al. (2021) to verify the effectiveness of our model.

### 4.1 IMPLEMENTATION DETAILS

The U-net backbone of SGMSong et al. (2021b) is copied and combined to build the two-stream network of the proposed method, refer to section 3.2. The input images are resized to $256 \times 256$ with normalization. In training, defect-free images were fed into a dual-stream denoising network, where the two streams learned $\epsilon_t$ and $\sigma_t^2$ for 1000 epochs, with time step range $T = 1000$. The loss function is defined in eq. (10), with a learning rate of $2e^{-5}$ for Adam optimizer. In testing, both defective and defect-free samples are fed into the trained model to obtain reconstructed images along with their corresponding variances. In ESM, the noise variance is manually set and follows the same schedule as in SGM, whereas in $ESM^{\sigma}$, the variance is predicted by the model. The time step $T_d$ for diffusion process is manually set for each class object, as different classes have different characteristics. For some object, e.g. capsule, even small $T_d = 50$ could achieve SOTA performance.

Consistent with previous work He et al. (2024), the area under the receiver operating characteristic(AUROC), is utilized for evaluating anomaly detection at both image and pixel levels. The ESM denotes the proposed ESM-based anomaly detection method, while $ESM^{\sigma}$ represents the ESM-based anomaly detection method with variance-guided diffusion.

Note that DiADHe et al. (2024) is a model for all categories. For fairness, its results are not included in the tables for comparison, and only visual results are provided. Since the results of Zhang et al. (2023b) and Lu et al. (2023) are both lower than GLADYao et al. (2024), neither of them is considered for comparison. For ADSPRShin et al. (2023b), it just detect the anomaly without reconstruction, which limit its application to other fields, such as image restoration. DDADMousakhan et al. (2024) is the most famous method with small model size, but its reconstruction process heavily depends on the input image, referring to $y_t - x_t$ in eq. (4), (5) and (6) of the DDAD paper. This will introduce uncorrected defects, leading to unsatisfactory reconstruction, and some false samples from DDAD paper are also shown in Appendix. For this reason, the result of DDAD is not included in our comparisons.

### 4.2 COMPARISON WITH STATE-OF-THE-ART METHODS

The comparison of results for MVTec-AD is shown in table 1, which contains the results of both discrete and continuous SOTA diffusion methods, along with SGMSong et al. (2021b) being the benchmark of continuous diffusion methods based on DSM. The proposed ESM (97.4/96.7) and $ESM^{\sigma}$ (98.7/97.2) methods outperform the continuous benchmark SGM (94.8/96.7) in AUROC at the image and pixel levels, approaching to that of GLAD(99.3/98.6). Considering the model size of GLAD is about 3 times larger than ours, the proposed methods' performance seem acceptable. The comparison of our variance-guided diffusion method with other approaches on the MVTec-AD Bergmann et al. (2019) dataset is shown in fig. 2.

In VisA dataset, the proposed methods, especially $ESM^{\sigma}$ could surpass GLAD(99.5/98.6), achieving the best performance(99.7/98.7) in both image-level and pixel-level AUROC, not to mention far more than the benchmark SGM, as shown in table 2.

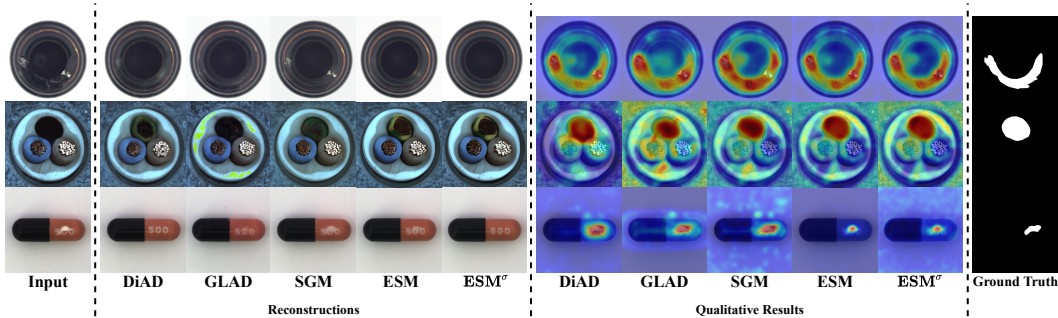

| **Input** | **DiAD** | **GLAD** | **SGM** | **ESM** | **ESM**$^\sigma$ | **DiAD** | **GLAD** | **SGM** | **ESM** | **ESM**$^\sigma$ | **Ground Truth** |
| | | **Reconstructions** | | | | | | **Qualitative Results** | | | |

Figure 2: The comparison of our methods with SOTA methods is shown in the figure. It is evident that our methods can achieve SOTA perform in both reconstruction quality and defect localization.

Table 1: The detection results are reported with image-level AUROC/pixel-level metrics on the MVTec-AD dataset. The best results across all methods are highlighted in bold.

| | Category | Non-Diffusion Method | | | Discrete Diffusion Method | Continuous Diffusion Method | | | |
| | | CutPaste | DRAEM | PatchCore | GLAD | ADSPR | SGM | ESM | ESM$^\sigma$ |
|---|---|---|---|---|---|---|---|---|---|
| Objects | Bottle | 98.2/97.6 | 99.2/**99.1** | 100./98.6 | 100./98.9 | 100./95.9 | 99.4/98.7 | 100./98.4 | **100.**/98.2 |
| | Cable | 81.2/97.6 | 91.8/94.7 | 99.5/**98.4** | **99.9**/98.1 | 94.2/ 96.9 | 99.2/97.0 | 99.0/98.2 | 99.8/96.4 |
| | Capsule | 98.2/97.4 | 98.5/94.3 | 98.1/**98.8** | **99.5**/98.5 | 97.2/96.6 | 93.4/92.4 | 85.3/92.2 | 94.1/95.6 |
| | Hazelnut | 98.3/97.3 | 100./**99.7** | 100./98.7 | **100.**/99.5 | 98.6/98.7 | 90.9/97.7 | 95.0/98.1 | 98.9/98.9 |
| | Metal Nut | 99.9/93.1 | 98.7/**99.5** | 100./98.4 | 100./98.4 | 96.6/ 96.6 | 98.0/95.4 | 99.9/91.6 | **100.**/94.9 |
| | Pill | 94.9/95.7 | **98.9**/97.6 | 96.7/97.1 | 98.1/97.9 | 96.1/**98.2** | 79.6/97.0 | 95.6/91.2 | 97.5/91.7 |
| | Screw | 88.7/96.7 | 93.9/97.6 | 98.1/99.4 | 96.9/99.1 | 98.6/99.5 | **99.4/99.6** | 98.8/99.5 | 98.8/99.5 |
| | Toothbrush | 99.4/98.1 | 100./98.1 | 100./98.7 | 100./**99.4** | 98.1/ 97.8 | 97.8/99.3 | 99.7/98.2 | **100.**/98.8 |
| | Transistor | 96.1/93.0 | 93.1/90.9 | 100./**96.3** | 98.3/96.2 | 98.7/94.7 | 91.2/88.0 | 94.5/95.5 | 93.6/95.2 |
| | Zipper | 99.9/**99.3** | 100./98.8 | 98.8/98.8 | 98.5/97.9 | 99.9/98.8 | 99.3/98.5 | 97.6/97.7 | 97.6/97.7 |
| Textures | Carpet | 93.9/98.3 | 97.0/95.5 | 98.7/98.9 | 99.0/98.5 | 91.7/ 96.4 | 99.5/99.4 | 98.5/99.3 | **99.8/99.4** |
| | Grid | 100./97.5 | 99.9/**99.7** | 98.2/98.7 | 100./99.6 | 100./98.9 | 100./99.3 | 100./99.1 | **100.**/99.0 |
| | Leather | 100./99.5 | 100./98.6 | 100./99.3 | 100./**99.8** | 99.9/ 99.3 | 100./99.4 | 100./99.5 | **100.**/99.3 |
| | Tile | 94.6/90.5 | 99.6/**99.2** | 98.7/95.6 | 100./98.7 | 99.8/96.8 | 99.9/98.5 | 99.4/98.2 | **100.**/98.2 |
| | Wood | 99.1/95.5 | 99.1/96.4 | 99.2/95.0 | 99.4/**98.4** | 96.1/95.4 | 73.7/90.7 | 97.9/94.0 | **100.**/95.0 |
| | Average | 96.1/96.0 | 98.0/97.3 | 99.1/98.1 | **99.3/98.6** | 97.7/97.4 | 94.8/96.7 | 97.4/ 96.7 | 98.7/97.2 |

Table 2: The detection results are reported with image-level AUROC/pixel-level AUROC metrics on the VisA dataset. The best results across all methods are highlighted in bold.

| Category | Non-Diffusion Method | | Discrete Diffusion Method | Continuous Diffusion Method | | |
| | DRAEM | PatchCore | GLAD | SGM | ESM | ESM$^\sigma$ |
|---|---|---|---|---|---|---|
| Candle | 89.6/91.0 | 98.7/99.2 | **99.9**/94.8 | 71.8/62.4 | 99.6/**99.4** | 98.0/99.0 |
| Capsules | 89.2/99.0 | 68.8/96.5 | 99.1/99.6 | 57.6/75.5 | 99.4/**99.7** | **99.6**/99.6 |
| Cashew | 88.3/85.0 | 97.7/**99.2** | 98.4/97.0 | 37.1/90.5 | 100./97.8 | **100.**/95.5 |
| Chewinggum | 96.4/97.7 | 99.1/98.9 | 99.6/99.1 | 96.4/98.0 | **99.9/99.6** | 99.6/99.4 |
| Fryum | 94.7/82.5 | 91.6/95.9 | 99.4/96.9 | 100./97.2 | 100./97.2 | **100.**/**97.2** |
| Macaroni1 | 93.9/99.4 | 90.1/98.5 | **99.9/99.8** | 86.9/97.2 | 99.8/98.0 | 99.8/98.6 |
| Macaroni2 | 88.3/99.7 | 63.4/93.5 | 98.9/**99.8** | 88.6/94.3 | 98.4/99.4 | **100.**/99.4 |
| Pcb1 | 84.7/98.4 | 96.0/**99.8** | 99.6/99.6 | 53.2/75.0 | 99.5/87.8 | **100.**/99.5 |
| Pcb2 | 96.2/94.0 | 95.1/98.4 | 100./ 98.6 | 70.3/37.8 | 99.7/97.9 | **100.**/98.4 |
| Pcb3 | 97.4/94.3 | 93.0/98.9 | 99.9/98.9 | 100./99.1 | 100./99.2 | **100.**/**99.2** |
| Pcb4 | 98.9/97.6 | 99.5/98.3 | 99.9/**99.5** | 83.3/81.1 | 99.9/99.1 | **100.**/99.4 |
| Pipe fryum | 94.7/65.8 | 99.0/99.3 | 98.9/99.4 | 91.2/96.8 | **100.**/**99.6** | 99.8/99.3 |
| Average | 92.4/92.0 | 91.0/98.1 | 99.5/98.6 | 78.0/83.7 | 99.7/97.9 | **99.7/98.7** |

The comparison between the proposed method and other SOTA methods on MPDD is shown in table 3, ESM$^\sigma$ (97.9/96.8) outperforms GLAD(97.5/98.7) in image-level AUROC, while lower in pixel-level AUROC.

As shown in table 4, let alone SOTA performance, the proposed method is more efficient than existing SOTA methods with smaller model size and fewer steps $T_d$. The selected values of $T_d$ are

Table 3: The detection results are reported with image-level AUROC/pixel-level AUROC metrics on the MPDD dataset. The best results across all methods are highlighted in bold.

| Category | Non-Diffusion Method | | Discrete Diffusion Method | Continuous Diffusion Method | | | |
| --- | --- | --- | --- | --- | --- | --- | --- |
| | DRAEM | PatchCore | GLAD | ADSPR | SGM | ESM | ESM$^\sigma$ |
| Bracket Black | 91.8/98.2 | 85.3/97.6 | 98.0/**99.4** | -/99.0 | 94.5/75.7 | 97.2/97.8 | **98.3**/97.5 |
| Bracket Brown | 90.3/63.7 | 92.5/**98.1** | 90.7/97.5 | -/97.7 | 98.3/36.9 | 98.0/95.4 | **99.6**/95.1 |
| Bracket White | 88.8/98.9 | 92.3/99.7 | **98.3/99.7** | -/86.2 | 91.4/37.5 | 91.3/97.1 | 91.7/97.2 |
| Connector | 100./91.2 | 100./99.4 | 100./98.2 | -/99.4 | 100./**99.7** | 99.8/92.4 | **100.**/92.6 |
| Metal Plate | 100./96.6 | 100./98.8 | 99.9/**99.4** | -/99.0 | 94.6/98.4 | 100./99.2 | **100.**/99.1 |
| Tubes | 94.7/95.9 | 77.4/97.2 | **98.1**/97.8 | -/**99.4** | 97.5/97.7 | 97.6/99.1 | 98.0/99.1 |
| Average | 94.3/90.7 | 91.3/98.5 | 97.5/**98.7** | -/96.8 | 96.1/74.3 | 97.3/96.8 | **97.9**/96.8 |

Table 4: Model Size and Diffusion Steps. Compared to other methods, our approach achieves higher efficiency.

| Method | Model Size | Timestep |
| --- | --- | --- |
| DiAD | 11.3GB | 1000 |
| GLAD | 3.2GB | 1000 |
| ESM | 0.98GB | $\leq 300$ |
| ESM$^\sigma$ | 0.98GB | $\leq 600$ |

detailed in the Appendix. ESM$^\sigma$ need twice steps of ESM. Because ESM$^\sigma$ requires recalculating the image variance at each diffusion step. Even though, ESM$^\sigma$ is faster than existing methods by 40%. Indeed, as $T_d$ varies from object to object, its average value is about 80 for ESM, i.e. more than 10 times faster compared to existing methods.

One core issue of this work is to investigate the distinctions between SGM and ESM. For comparison, the denoising processes of SGM and ESM for one defective sample are shown in fig. 3(more sample could be found in the Appendix). The intermediate results and corresponding variance range are given at $t = 200, 150, 100, 80, 50, 20, 0$. The variance for SGM is $\sqrt{1 - \bar{\alpha}_t}$, while the variance of ESM is provided by neural network, denoted by $(min(\sigma_t^2), max(\sigma_t^2))$, refer to eq. (3) and eq. (4). Here, the lower bound $min(\sigma_t^2)$ of the ESM variance is set as 0.01 to avoid being divided by a too small abnormal value. fig. 3 shows that the denoising process of ESM reconstructing the samples at about $t = 100$ versus $t = 20$ for SGM (indicated by the red boxes), demonstrating higher efficiency, which has the potential of early-stop in future research, with the range of ESM's variance, i.e. the value of $max(\sigma_t^2)$, being stable. In comparison, the variance of SGM changes from 0.341 to $10^{-4}$, making the result of denoising process depends on the last stage $t = 50$ to $t = 0$, as variance appearing in the denominator of eq. (4). Of course, this will not be a problem for the variance-guided diffusion model, since the variance appearing in the numerator, refer to eq. (13).

### 4.3 ABLATION STUDY

The difference between the benchmark SGM and the proposed methods have been given out in above tables. Also, the difference between ESM and its variance-guide version ESM$^\sigma$ could be found in section 4.2. In this section, the ablation experiments for the four anomaly detectors are conducted in challenging dataset MPDD using ESM$^\sigma$, and the results are shown in table 5. It could be found that all detectors should be utilized to improve the performance. Notice that detectors for $S_{KL}$ and $S_U$ depend on the predicted variance $\sigma_t^2$, which could only be provided by ESM or ESM$^\sigma$.

Table 5: The detection results of ESM$^\sigma$ are reported with image-level AUROC/pixel-level AUROC metrics on the MPDD dataset. The best results are highlighted in bold.

| $S_{Cos}$ | $S_{RE}$ | $S_U$ | $S_{KL}$ | Average |
| --- | --- | --- | --- | --- |
| ✓ | ✓ | | | 93.8/93.5 |
| ✓ | ✓ | ✓ | | 92.3/95.5 |
| ✓ | ✓ | | ✓ | 93.9/93.5 |
| ✓ | ✓ | ✓ | ✓ | **97.9/96.8** |

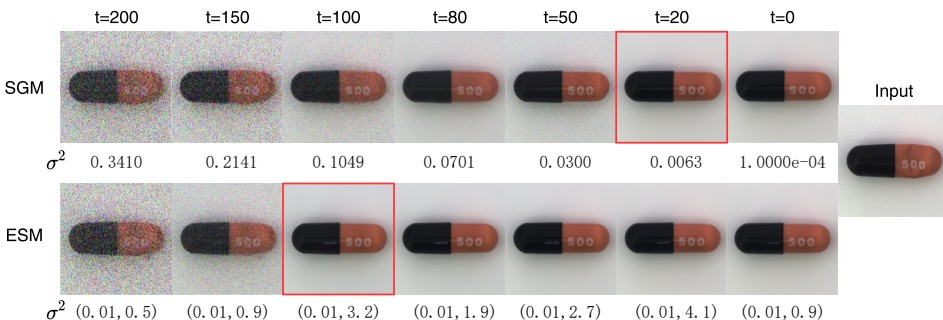

Figure 3: The denoising process of SGMSong et al. (2021b) vs ESM. Intermediate denoising results are displayed with their corresponding variance under each image, at $t = 200, 150, 100, 80, 50, 20, 0$. It is easy to find that the denoising process of ESM is more efficient than SGM, with the corresponding variance being stable.

## 5 DISCUSSION

In this work, it is assumed that the characteristic vector $X_t$ (including the input image $X_0$) follows the Gaussian distribution with diagonal covariancesMao et al. (2020), which usually does not hold for common images. This limitation can be solved by employing an image encoder to project images into feature vectors, which could be considered following the Gaussian distribution with diagonal covariance, as the same as that in DiADHe et al. (2024). In the other side, the images are usually generated from Gaussian noise, by repeatedly adding Gaussian noise(refer to DDPM), and therefore the final generated images are considered following Gaussian distribution.

## 6 CONCLUSION

As demonstrated by Song et al. (2021b) (ICLR 2021 Outstanding Paper Award), discrete diffusion model is equivalent to continuous diffusion model. However, some fundamental issues remain under-explored, one of which is the usage of DSM(denoising score match) instead of ESM(explicit score matching), as ESM could not be obtained by modern diffusion models. Thus, in this work, we proposed a variance-guided diffusion method that could estimate the true probability score (i.e., ESM) while achieving SOTA performance with high efficiency in defect detection task. Also, the difference between ESM and DSM is investigated, and the experiment results show that there would exist risk of performance downgrade in both accuracy and efficiency by using DSM instead of ESM, which unfortunately is the most common way for recent diffusion models. Furthermore, the proposed framework and methods(ESM and ESM$^\sigma$) could be easily applied to nearly all existing diffusion models, such as stable diffusion Rombach et al. (2022) or even Sora etc. for kinds of tasks, e.g. image generation Zhang et al. (2023a); Song et al. (2021a).

## REPRODUCIBILITY STATEMENT

We have made every effort to ensure the reproducibility of our work. All datasets used in our experiments are publicly available and explicitly cited in the paper. Comprehensive details on data preprocessing, model architectures, training procedures, evaluation protocols, and mathematical derivations are provided in the main text and appendix. We report all key hyperparameters, optimization settings, and stopping criteria. To mitigate randomness, each experiment was repeated with multiple random seeds, and average results are reported. We will release the source code, trained models, and detailed instructions for reproducing our results upon publication.

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

# A APPENDIX

## A.1 THE DERIVE OF $\beta(t)$ FOR VARIANCE-GUIDE DIFFUSION

For variance guide method, shown in eq. 11, the variance could be calculated as follows under Gaussian distribution assumption: define a small time interval $\triangle t$, we have

$$X_{t+\triangle t} - X_t = -\frac{\sigma_t^2}{2} X_t \triangle t + \sigma_t \triangle W \tag{16}$$

and by taking expectation,

$$E\{X_{t+\triangle t}\} - E\{X_t\} = E\{-\frac{\sigma_t^2}{2} X_t \triangle t\} + \sigma_t E\{\triangle W\}$$

$$\Rightarrow \mu_{t+\triangle t} - \mu_t = -\frac{\sigma_t^2}{2} \mu_t \triangle t \tag{17}$$

$$\Rightarrow \mu_{t+\triangle t} = (1 - \frac{\sigma_t^2}{2} \triangle t)\mu_t$$

Then, we have

$$\sigma_{t+\triangle t}^2 = E\{(X_{t+\triangle t} - \mu_{t+\triangle t})^2\}$$

$$= E\{X_{t+\triangle t}^2\} - \mu_{t+\triangle t}^2 \tag{18}$$

$$= E\{(X_t - \frac{\sigma_t^2}{2} X_t \triangle t + \sigma_t \triangle W)^2\} - \mu_{t+\triangle t}^2$$

With $E\{(\triangle W)^2\} = \triangle t$, $E\{\triangle W\} = 0$ and $X_t$ being independent with $\triangle W$, the above equation turns into

$$\sigma_{t+\triangle t}^2 = E\{X_t^2\} + \frac{\sigma_t^4}{4}(\triangle t)^2 E\{X_t^2\} + \sigma_t^2 E\{(\triangle W)^2\} - \sigma_t^2 E\{X_t^2\}\triangle t - \mu_{t+\triangle t}^2$$

$$= \sigma_t^2 + \mu_t^2 + \frac{\sigma_t^4}{4}(\triangle t)^2 E\{X_t^2\} + \sigma_t^2 \triangle t - \sigma_t^2(\sigma_t^2 + \mu_t^2)\triangle t - (1 - \frac{\sigma_t^2}{2}\triangle t)^2 \mu_t^2$$

$$= \sigma_t^2 + \mu_t^2 + \sigma_t^2 \triangle t - \sigma_t^2(\sigma_t^2 + \mu_t^2)\triangle t + o(\triangle t) - (1 - \sigma_t^2 \triangle t)\mu_t^2 \tag{19}$$

$$= \sigma_t^2 + \sigma_t^2 \triangle t - \sigma_t^4 \triangle t + o(\triangle t)$$

$$\Rightarrow \sigma_{t+\triangle t}^2 - \sigma_t^2 = \sigma_t^2 \triangle t - \sigma_t^4 \triangle t + o(\triangle t)$$

where $o(\triangle t)$ is the high order element of $\triangle t$. Let $\triangle t \to 0$, we get

$$2\sigma_t d\sigma_t = -\sigma_t^4 dt + \sigma_t^2 dt \tag{20}$$

with initial $\sigma_0$ as the boundary condition. Then $\sigma_t$ could be obtained from the above differential function. Thus, $\sigma_t$ is just a function of time $t$ and $\sigma_0$, not related to random variable $X$.

## A.2 MORE EXPERIMENT RESULTS

One more example of the comparison between SGM and ESM during the denoising process is shown in fig. 4. The corresponding diffusion steps $T_d$ for all dataset categories are listed in tables.

Also, fig. 5 and fig. 6 provide more visual comparisons between our methods and other state-of-the-art approaches. The results demonstrate that our methods achieve high reconstruction fidelity and defect localization performance. fig. 7 illustrates the reconstruction performance of DDAD, which demonstrates suboptimal restoration results.

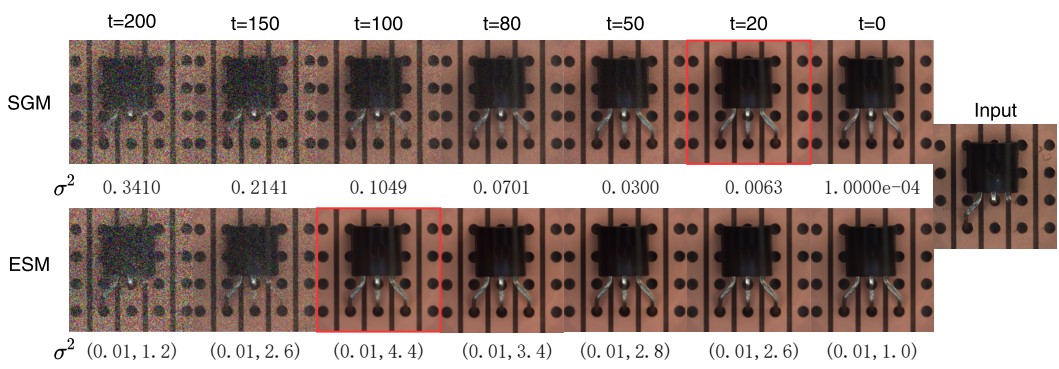

Figure 4: The denoising process of SGM vs ESM on the transistor. Intermediate denoising results are displayed with their corresponding variance under each image, at $t = 200, 150, 100, 80, 50, 20, 0$.

Table 6: Categories and Corresponding $T_d$ Values in the MVTec-AD Dataset.

| Category | $T_d$ | Category | $T_d$ | Category | $T_d$ |
|---|---|---|---|---|---|
| Bottle | 100 | Pill | 164 | Carpet | 250 |
| Cable | 200 | Screw | 45 | Grid | 100 |
| Capsule | 48 | Toothbrush | 200 | Leather | 49 |
| Hazelnut | 100 | Transistor | 100 | Tile | 200 |
| Metal Nut | 100 | Zipper | 280 | Wood | 20 |

Table 7: Categories and Corresponding $T_d$ Values in the VisA Dataset.

| Category | $T_d$ | Category | $T_d$ | Category | $T_d$ |
|---|---|---|---|---|---|
| candle | 3 | Fryum | 50 | Pcb2 | 50 |
| Capsules | 50 | Macaroni1 | 50 | Pcb3 | 50 |
| Cashew | 50 | Macaroni2 | 50 | Pcb4 | 50 |
| Chewinggum | 10 | Pcb1 | 50 | Pipe fryum | 50 |

Table 8: Categories and Corresponding $T_d$ Values in the MPDD Dataset.

| Category | $T_d$ | Category | $T_d$ | Category | $T_d$ |
|---|---|---|---|---|---|
| bracket black | 2 | bracket brown | 3 | bracket white | 2 |
| connector | 3 | metal plate | 50 | tubes | 1 |

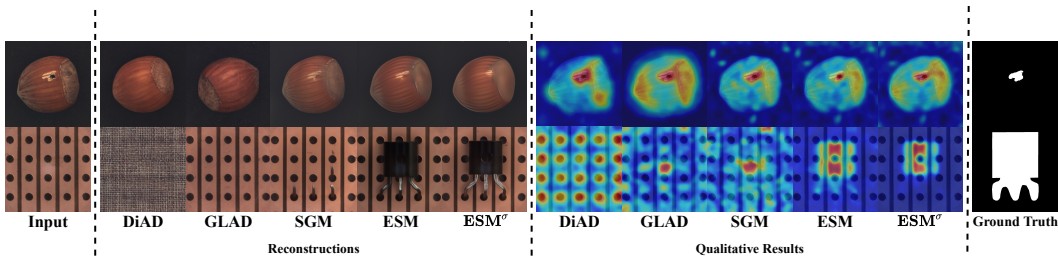

Figure 5: The comparison of our methods with SOTA approaches on the MVTec-AD dataset is shown in the figure. It is evident that our methods can achieve SOTA perform in both reconstruction quality and defect localization.

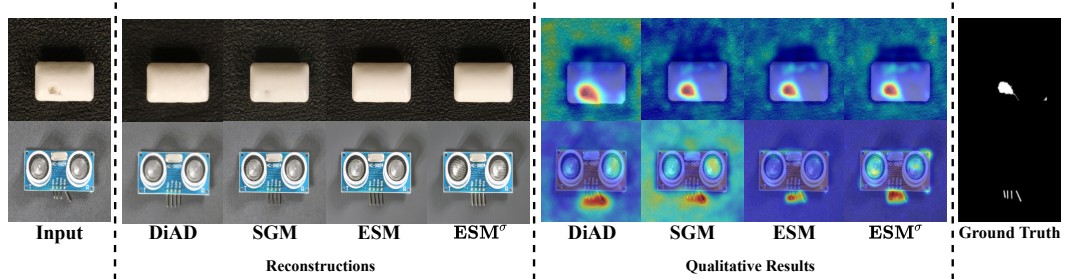

Figure 6: The comparison of our methods with SOTA approaches on the VisA dataset is shown in the figure. It is evident that our methods can achieve SOTA perform in both reconstruction quality and defect localization.

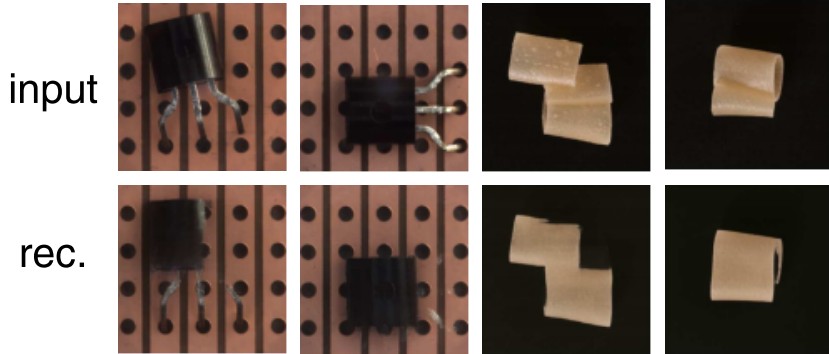

Figure 7: The reconstructed images of DDAD reveal that its effectiveness in defect restoration is limited.

