# OpenReview forum: "Continuous Diffusion Models with Explicit Score Matching for Highly Efficient Anomaly Detection"
_ICLR.cc/2026/Conference — Submitted to ICLR 2026_

### Official Review · Reviewer_EWrT · 2025-10-14

**Soundness:** 1
**Presentation:** 1
**Contribution:** 2
**Rating:** 0
**Confidence:** 5

**Summary:**

The paper proposes a diffusion-based anomaly detection framework which the authors refer to as a ``continuous-time explicit score matching (ESM)’’ model, but which is in practice implemented as a discretized DDPM-style denoising network with dual prediction heads for the mean and variance of a Gaussian noise model. The idea is to train these heads by maximum likelihood so that the model can compute an “explicit” marginal score, contrasting this with the denoising score used in standard continuous diffusion training. The authors argue theoretically that DSM coincides with ESM only for the squared loss case, to motivate their design and a variance-guided diffusion procedure that replaces the fixed schedule with the predicted variance for greater stability and fewer steps. At inference, the system reconstructs images and aggregates several residual- and feature-based anomaly scores, reporting strong accuracy with smaller models and reduced sampling. The paper also compares against other methods on MVTec-AD, VisA, and MPDD.

**Strengths:**

- Good industrial anomaly detection motivation and use of standard datasets.
- Simple dual-head design ($\mu$, $\sigma^2$) trained via negative log-likelihood (NLL) is easy to implement.
- The paper runs broad experiments and explores multiple anomaly scores showing good performance.

**Weaknesses:**

- Continuous vs discrete diffusion usually refers to the type of data used in training, the authors are more specifically talking about discrete-time or continuous-time diffusion model. This should be fixed in the title and throughout the paper. The paper writes the correct reverse SDE (Eq. 2) and then “omits the random noise” to present an ODE version (Eq. 5). There is no justification or reference for this removal; it changes the generative semantics and is not the standard Song et al. reverse-time SDE/ODE derivation. This does not make the method a continuous-time diffusion model and the actual method is a discretized DDPM-style sampler with fixed $T_d$, thus, it is not a continuous-time score-based model.

- The paper equates explicit score matching with the identity $\nabla_{x}\log p_t(x)=-(x-\mu_t)/\sigma_t^2$ and calls this “the formula of ESM” (Eq. 3). But this is simply the Gaussian score; it only holds if the marginal p_t(x) is Gaussian, which is not true in general for diffusion marginals (they are data distributions convolved with Gaussians). The expression corresponds instead to the conditional Gaussian score used in denoising score matching (DSM) under the Variance Preserving (VP) diffusion process. True ESM (Hyvärinen, 2005) involves matching the marginal score using the divergence-based objective.

- Eq. (6) defines ESM as matching $s_\theta$ to the marginal score $\nabla\log p_t(x)$ inside an $\ell^l$ loss. This assumes pointwise access to the true marginal score, which is precisely what Hyvärinen’s ESM avoids via the divergence trick. The paper later cites a Vincent-style equivalence for l=2 (Eq. 12), but the preceding derivation is still built on an intractable oracle quantity.

- The paper “expands” $\|s_\theta-\nabla\log p_t\|^l$ using binomial-style sums with inner products of exponentiated vectors (e.g., $\langle s^{l-k}$, ($\nabla\log p)^k\rangle$). That expansion is not a valid identity for vector norms; it implicitly treats the vector norm to a power as if it were a scalar binomial, which is wrong. These steps underlie Eq. (8) and Eq. (9), so the subsequent comparison built on them is unreliable.

- Replacing $\beta(t)$ with a predicted, input-dependent $\sigma^2(x,t)$ (Eqs. 16–17) breaks the standard SDE setup where drift/diffusion are time-dependent (or parameter-dependent) but not input-dependent in that way. It’s unclear what distribution is simulated, and arguments are missing.

- Implementation details admit $T_d$ is “manually set for each class,” with tables of hand-picked $T_d$ for each dataset category. This amounts to selecting hyperparameters directly on the test set, effectively overfitting and undermining the validity of their comparisons to baselines that use fixed or globally defined schedules.

- A substantial part of the final score mixes four detectors (cosine/DINO, uncertainty-weighted residual, KL, etc.). It’s unclear how much of the improvement comes from the dual-stream diffusion versus these add-ons. An ablation (Table 5) exists but not every component is individually analyzed. Would a simple diffusion model work as well using this assemble of four detectors?

- The writing quality is poor and often confusing; the flow between sentences is abrupt and lacks coherence. Informal phrasing (e.g. “actually”) appears throughout, which undermines academic tone. Several citations are incorrectly formatted, missing parentheses or proper spacing before the reference marker. Conclusions are often stated without logical grounding in the preceding arguments, creating a sense of unjustified claims rather than reasoned progression.

- The provided supplemental code is poorly structured and difficult to follow; its quality does not meet common research reproducibility or clarity standards.

In summary, the method used is still a discrete-time diffusion model using DSM, which goes against the whole argument of the paper.

**Questions:**

- $T_d$ was set per class. How was this chosen without test leakage?
- As you claimed an efficient method, what are the gains in training and inference speed compared to other methods beyond the number of timesteps and model size?
- How exactly are the images reconstructed? Is there a forward noising process first? I see it is the case in the code, but it is not explained in the paper.
- What happens if the variance predicted by the model is always zero (or close to zero)? It looks like the model would become a standard DDPM. I suspect this is the case.

Please see weaknesses.

---

> ### Author Response · Authors · 2025-11-27
>
> Weaknesses:
> 1 The SDE/ODE formulations differ only in the sampling path, and this distinction is part of the work by Song et al. Our ESM framework supports both SDE and ODE formulations, and we adopt the ODE version in experiments solely for simplicity.We derive ESM from the continuous formulation and implement it using $\nabla \log p_t(x)$, which comes from continuous-time diffusion models and is generally absent in discrete DDPMs. Moreover, Song et al. have shown that continuous and discrete diffusion models are equivalent. Therefore, our method can be viewed either as a continuous-time diffusion model or as an improvement and generalization of DDPM discrete methods, without affecting the contributions of this work.
>
> 2 For the non-Gaussian case, the discussion section notes that one may follow approaches similar to DIAD by projecting the data distribution into a feature space where it becomes approximately Gaussian, as also seen in VAE or GLAD. At the same time, most diffusion-model–generated images already lie in a Gaussian space because the process starts from Gaussian noise and adds independent Gaussian perturbations at each step, and this point has been included in the discussion section. Even under the Gaussian assumption, existing DSM methods are still unable to compute the true score. By contrast, our work introduces maximum likelihood as the loss in a principled way, eliminating the need for the marginal score used in divergence-based objectives.
> 3 Since the Gaussian distribution is adopted, the ESM becomes computable.
>
> 4 This part has been corrected, and the relevant equations are now given in Eq. (6)–(13). The conclusion remains unchanged: in general, DSM is not equivalent to ESM.
>
> 5 Under the assumptions of a Gaussian distribution and independent Gaussian noise, the variance $\sigma_t$ is analytically computable and becomes a deterministic function of t. This function comes from the truth that the variance $\sigma^2_t$ at time $t$ could be estimated from the initial variance $sigma^2_0$ of the input and independent noise, under Gaussian assumption. Thus, using $\beta(t) =\sigma_t$ does not change the form of the equations because the process still satisfies the SDE in practice (see eq.(18) and the corresponding equations (22)-(26)in the Appendix). In implementation, we let the network predict $\sigma_t$​ instead of directly using the closed-form solution to account for potential errors such as those arising from $\sigma_0$. Further investigation will be conducted in future work.
>
> 6 This parameter is selected using a validation set, since the diffusion model for generation is trained only on normal samples, while abundant anomalous samples are available for training and validating the detector. Moreover, the choice of T_d​ is determined simply by evaluating how well the anomalous samples in the training set are restored.
> 7 Uncertainty-weighted residual and KL terms are based on the predicted variance, which cannot be obtained from a standard diffusion model.
> 8We have improved the description.
>
> 9 The full code will be released upon acceptance.
> Question:
> 1 A validation set is used, where the diffusion model is trained only on normal samples, and all anomalous samples are reserved for validation.
> 2 Our method achieves SOTA detection primarily under settings with relatively small time steps and model size.
> 3 Except for the final variance-guided noise injection, the remaining noise injection follows existing methods, i.e., one-step noise addition.
> 4 During maximum likelihood training, the variance appears in the denominator(eq.(15) and eq.(16)), so the model naturally avoids extremely small values. If the variance is very small, it generally indicates minimal noise and limited denoising effect; we set a lower bound for the variance. As shown in Fig. 3, the variance remains within a certain range, particularly when using our proposed variance-guided noise injection, which effectively moves the variance term to the numerator(eq.(19)).

---

> ### Comment · Reviewer_EWrT · 2025-11-27
>
> I thank the authors for their detailed response. However, the rebuttal actually confirms the fundamental concerns raised in the initial review regarding both the theoretical soundness and the experimental validity of this work.
>
> **Misrepresentation of Score Matching Theory and Mathematical Errors**. The paper claims DSM is a degraded approximation of ESM. This is incorrect as Vincent (2011) proved that Denoising Score Matching is mathematically equivalent to Explicit Score Matching. Your proposed method is actually inferior because it replaces this unbiased estimator with a biased Maximum Likelihood Estimation using a parametric Gaussian proxy.
>
> **The "Gaussian Fallacy" Invalidates the "Explicit Score Matching" Claim**. In the rebuttal, you state that "Since the Gaussian distribution is adopted, the ESM becomes computable" and argue that diffusion-generated images already lie in a Gaussian space. This reveals a critical misunderstanding of diffusion models. The marginal distribution $p_t(x)$ is the convolution of the data distribution $p_{data}(x)$ with a Gaussian kernel; for image data, this results in a highly multimodal infinite Gaussian mixture, not a single unimodal Gaussian. By strictly defining the score via Equation 3 ($-\frac{\tilde{x} - \mu_t}{\sigma_t^2}$) and optimizing it via Maximum Likelihood (Equation 15), you mathematically force the model to approximate this complex mixture as a single Gaussian blob. Consequently, your "explicit score" points merely toward the global average of the dataset rather than the data manifold, effectively reducing the model to a unimodal Denoising Autoencoder rather than a generative diffusion model (your loss in equation 14 is merely the negative log-likelihood of a Gaussian).
>
> **Contradiction in the Reconstruction Formulation** You argue that your method is superior because it targets the marginal $p_t(\tilde{x})$ rather than the conditional $p_t(\tilde{x}|x)$. However, the anomaly detection task (reconstruction) is inherently conditional: you must recover $x_0$ given a specific $x_t$. If your model truly learned only the marginal distribution (which you define as a single Gaussian), it would lack the conditional information necessary to map a specific noisy input back to its clean counterpart. The fact that it works implies that you just implemented DSM with a slightly modified, unsound, loss.
>
> **The "Continuous-Time" Framing is Misleading**. The paper heavily emphasizes the "continuous-time" aspect to distinguish the method from standard approaches, yet your implementation utilizes a standard discrete-time scheduler. In the rebuttal, you admit that continuous and discrete models are equivalent (citing Song et al.). If the formulations are equivalent and your implementation is discrete, framing the contribution around "continuous diffusion" creates a false distinction. It provides no methodological advantage over standard discrete baselines and contradicts the implementation details.
>
> **Confirmed Data Leakage**. Your use of a validation set containing "abundant anomalous samples" to tune class-specific parameters ($T_d$) is a violation of the unsupervised anomaly detection protocol. This constitutes data leakage, rendering any comparison against unsupervised baselines (which do not access anomalies during tuning) invalid.
>
> Finally, although you claim that the derivation has been corrected, the manuscript remains unchanged and the proof still contains fundamental errors. Either way, as mentionned above, there is a fundamental misunderstanding of ESM and DSM, which leads to a useless difference analysis. I also note that, despite emphasizing efficiency, the released code uses two separate U-Nets for the mean and the variance, which is highly inefficient compared to prior work that simply increases the number of channels when predicting variance.
>
> I am remain strongly toward a rejection, this paper needs a whole rework both on the theoretical and experimental side.

---

> > ### Author Response · Authors · 2025-11-29
> >
> > Thank you for your prompt comments.
> > We apologize for the delay in updating the manuscript and kindly ask the reviewers to consider the revised version. In the revision, we have corrected the DSM and ESM formulations and included the derivation of the variance-guided diffusion process.
> >
> > The reviewer argues that the paper misinterprets score-matching theory, incorrectly claims that DSM is inferior to ESM, and instead proposes a biased Gaussian-based MLE that is theoretically weaker than the unbiased DSM estimator; however, Vincent (2011) establishes equivalence only for the special case of l=2, as shown in our revised Eqs. (6)–(7), and the learned DSM and ESM objectives remain fundamentally different, which is consistent with the original ESM formulation in Eq. (2).
> >
> > Regarding the reviewer’s concerns that $p_t(x)$ forms an infinite Gaussian mixture rather than a single Gaussian, and that optimizing the score via Eq. 3 and the MLE loss in Eq. 15 would collapse this mixture into one unimodal Gaussian centered on the dataset mean—thus reducing the method to a unimodal denoising autoencoder—our approach follows the same principle used in MAE and VAE. In practice, the generative results show that this assumption is sufficient. Moreover, the reviewer’s interpretation is not accurate: the model does not force all samples to share one Gaussian. Each input, such as $X_1$​ and $X_2$​, corresponds to its own Gaussian distribution, e.g., $N(\mu_1, \sigma_1)$and $N(\mu_2, \sigma_2)$, rather than a single shared Gaussian.
> >
> > The reviewer argues that claiming to model the marginal rather than the conditional is contradictory, because anomaly reconstruction is inherently conditional, and if the model truly learned only a marginal Gaussian it could not map$ x_t​ $back to $x_0$​, suggesting that the method is effectively just DSM with an unsoundly modified loss. In response, because our choice of $T_d​$ is much smaller than that used in other approaches, the noisy input still retains sufficient information for accurate reconstruction. In contrast, a conditional formulation would introduce sample-specific information together with potential abnormal features, which is precisely why we avoid adopting a conditional scheme.
> >
> >
> > The reviewer argues that the paper’s emphasis on “continuous-time” diffusion is misleading because the implementation ultimately uses a discrete-time scheduler, and if continuous and discrete formulations are equivalent—as acknowledged in the rebuttal—then framing the contribution around “continuous diffusion” creates a false distinction and offers no practical advantage over standard discrete baselines. In response, our method in fact directly follows the governing equation presented in Eq. (13); the implementation simply uses a small discrete step size dtdtdt as a numerical approximation. This same formulation can also be solved with an ordinary differential equation solver, which would realize the fully “continuous” version, meaning that the underlying method remains continuous by formulation, with discretization appearing only as a computational approximation.
> >
> > The reviewer claims that data leakage has occurred.Anomalous samples are used only to select hyperparameters such as $T_d​$ and coefficients; information about defect types is not utilized. Since the test set is not involved in hyperparameter selection, this does not constitute test data leakage.
> >
> > Finally, regarding the previously raised concern that “replacing \beta(t) with a predicted, input-dependent $\sigma^2(x,t)$ (Eqs. 16–17) breaks the standard SDE setup,” please refer to Eqs. (16)–(20).We have also addressed the issue of very small variance.Regarding the reviewer’s questions on efficiency and reconstruction: our method achieves SOTA detection primarily under settings with relatively small timesteps and model size. As for image reconstruction, except for the final variance-guided noise injection, all other noise injections follow existing methods, i.e., one-step noise addition. All previously raised concerns on these points have been addressed.

---

### Official Review · Reviewer_Z6vw · 2025-10-28

**Soundness:** 2
**Presentation:** 2
**Contribution:** 2
**Rating:** 4
**Confidence:** 2

**Summary:**

This work aims to improve diffusion-based anomaly detection by explicitly learning the score function via a maximum-likelihood dual-stream network. The proposed approach replaces the denoising score with a direct estimate of the log-likelihood gradient and integrates a variance-guided diffusion process to enhance efficiency.

**Strengths:**

- Clear motivation to bridge DSM and ESM formulations in diffusion modeling.
- Extensive experimental evaluation across multiple datasets.

**Weaknesses:**

- The ESM vs. DSM comparison is largely restating prior work (Vincent, 2011) and does not offer new theoretical insights.
- Improvements over baselines are marginal (1–2%) and within variance; the claim of “SOTA” performance is somehow overstated.
- No systematic exploration of the role of the number of diffusion steps, or parameter choices.

**Questions:**

N/A

---

> ### Author Response · Authors · 2025-11-27
>
> Regarding the comment that our ESM vs. DSM comparison largely restates prior work (Vincent, 2011) without offering new theoretical insights:
> our main focus, contribution, and innovation lie in resolving the computational challenges of ESM, including the maximum-likelihood–based training objective, which can be viewed as a more general form of the training objective used in conventional diffusion models, as well as the corresponding algorithmic framework and the subsequently developed variance-based noise-injection strategy. Vincent’s work only provides the theoretical basis for the performance gap between DSM and ESM (for the l=2 case; the other cases are shown in the revised manuscript in Eq. (6)–(13)). Our experiments show that adopting ESM leads to performance improvements and more stable variance, as illustrated in Fig. 3 and in the variance range displayed below the figure.
> Regarding the statement that improvements over baselines are marginal (1–2%), within variance, and that the claim of “SOTA” performance is overstated:
> our claim of SOTA performance refers to the combined advantages in both accuracy and efficiency, including model size. A well-known limitation of existing diffusion models is their slow inference speed, and our method effectively improves this aspect.
> Regarding the comment that there is no systematic exploration of the number of diffusion steps or parameter choices:
> these parameters are selected based on the validation set. We emphasize that our experimental procedure follows the settings of the compared methods without introducing special modifications. Since the diffusion model used for reconstruction is trained only on normal samples, we have sufficient abnormal samples to serve as training and validation data for the detector. Moreover, the primary focus of this work is ESM itself, while anomaly detection serves only as a downstream task to demonstrate that the proposed ESM-based method can produce high-quality reconstructions and generated samples.

---

### Official Review · Reviewer_RhCF · 2025-11-01

**Soundness:** 3
**Presentation:** 2
**Contribution:** 3
**Rating:** 8
**Confidence:** 4

**Summary:**

This paper investigates the use of explicit score matching (ESM) as an alternative to the widely used denoising score matching (DSM) in the context of anomaly detection.

The authors consider the anomaly detection task by computing an anomaly score S via reconstruction (i.e. comparing the input image with its projection into the “normal” domain as by some model) and some additional terms. They first rederive the cases where ESM and DSM are or are not equivalent and motivate the proposed estimation of the variance and the corresponding loss function for the ESM. For the experiments they use standard benchmark datasets like MVTec-AD etc, and show  the performance of the proposed model, but also relevant baselines.

- introduce explicit score matching vs classical Denoising Score Matching
- “Also, the difference between ESM and DSM is investigated, and the experiment results show that there would exist risk of performance downgrade in both accuracy and efficiency by using DSM instead of ESM, which unfortunately is the most common way for recent diffusion models.”
- “However, some fundamental issues remain under-explored, one of which is the usage of DSM(denoising score match) instead of ESM(explicit score matching), as ESM could not be obtained by modern diffusion model”

**Strengths:**

The authors show that their proposed continuous diffusion model with variance guided ESM outperform the various baselines and that ESM with variance guidance is worth investigating further. They show the performance across multiple standard benchmark datasets where their proposed method shows an impressive performance. The authors promised to publish their code, which is necessary for this work to be reproducible. The authors also show that the proposed method is also more efficient in terms of model size and compute, compared to the baselines.

**Weaknesses:**

In the method section (3.1) the authors show that for l>2 the terms for ESM and DSM are not equivalent, however there is no motivation why we would even consider anything other than l=2 in the first place. This would in my opinion be necessary to motivate the method, and it should also be reported what “l” is being used in the end.

The anomaly score is composed of various terms, among them the cosine similarity of DINO features, which do not directly seem to have any connection to the proposed method. It would be nice to also show the same ablation but without these, to see what part contributes how much.

In the method section the authors rederive the difference/equivalence between ESM and DSM which follows the work of Vincent as cited. While this context is appreciated for the motivation, it contains (along with the rest of the paper) some typos and inconsistencies that make it hard to follow. For instance, the jump from Eq8/9 to Eq 10/11 does not explain why the absolute value within the expectation can just be ignored. While true for even “l”, it is unclear in general.

**Questions:**

In the method section (3.1) the authors show that for l>2 the terms for ESM and DSM are not equivalent, however there is no motivation why we would even consider anything other than l=2 in the first place. This would in my opinion be necessary to motivate the method, and it should also be reported what “l” is being used in the end.

The anomaly score is composed of various terms, among them the cosine similarity of DINO features, which do not directly seem to have any connection to the proposed method. It would be nice to also show the same ablation but without these, to see what part contributes how much.

In the method section the authors rederive the difference/equivalence between ESM and DSM which follows the work of Vincent as cited. While this context is appreciated for the motivation, it contains (along with the rest of the paper) some typos and inconsistencies that make it hard to follow. For instance, the jump from Eq8/9 to Eq 10/11 does not explain why the absolute value within the expectation can just be ignored. While true for even “l”, it is unclear in general.

---

> ### Author Response · Authors · 2025-11-27
>
> Regarding the motivation for considering l>2 and clarifying which value of l is ultimately used, our discussion of the l>2 case is not intended to imply that the final model employs an l>2 loss. Instead, the goal is to demonstrate that ESM and DSM become strictly nonequivalent when l>2, thereby showing that the commonly used DSM formulation possesses a special approximation property only at l=2. We provide this proof solely to theoretically highlight the difference between DSM and ESM, and we further verify experimentally that existing methods based on DSM indeed sacrifice performance. During training, our method adopts maximum likelihood, and during denoising we directly compute the gradient under the Gaussian assumption rather than relying on indirect approximations (e.g., DSM with l=2). Therefore, no approximation with respect to l is required, or equivalently, the method can be viewed as valid for any l, since under a Gaussian distribution our approach enables direct computation of the score.
> Regarding the composition of the anomaly score and the role of the DINO feature cosine similarity, the inclusion of DINO feature cosine similarity follows standard practice in anomaly detection. Its role is not to replace diffusion-based reconstruction but to provide complementary cues, consistent with the design of leading diffusion-based anomaly detection approaches such as GLAD and ADSPR. Our main innovation focuses on the generative (i.e., reconstruction) component, as achieving SOTA performance requires both reconstruction and detection to be as strong as possible, while the remaining parts largely follow existing methods. In addition, since our method predicts the variance, we introduce two additional detection heads, $S_u​$ and $S_{\mathrm{KL}}$​, derived from the variance.
> Regarding the derivation of the ESM–DSM relationship and the handling of the absolute value when transitioning from Eq. 8/9 to Eq. 10/11, we appreciate the reviewer’s attention and have improved the completeness and clarity of the derivation in the revised manuscript, including Eq. 6–13. This part of the equations has been corrected, and for cases where $l \neq 2$, the result is straightforward to verify by directly substituting two Gaussian distributions.
> Overall, the analysis of the l>2 case establishes the boundary conditions of DSM’s approximation, the DINO feature term is an industry-standard discriminator independent of the diffusion reconstruction process, and the notational issues in the equations have been carefully addressed.

---

### Official Review · Reviewer_QycL · 2025-11-05

**Soundness:** 2
**Presentation:** 1
**Contribution:** 2
**Rating:** 0
**Confidence:** 4

**Summary:**

The goal of this paper is to introduce a diffusion model by explicit score matching. This is claimed to be achieved by using a dual stream encoder-decoder to calculate the expected value and variance of the noise at each time-step, thereby directly estimating the noise density without the need to calculate the conditional score via denoising score matching. However, the paper is very poorly written and there are many flaws in the logic, ambiguities in the theoretical framework, and lack of proper experimental validation that render the paper unsuitable for acceptance.

**Strengths:**

The paper presents an interesting approach in score matching for diffusion models, particularly the denoising (backward) process.

**Weaknesses:**

The paper has many flaws that need to be addressed for a proper evaluation of the proposed framework. These include improvements in the theory, originality, significance, clarity, and experimental design. Please see more details below:

- From a theoretical framework, the purported claim that current diffusion models have to use the conditional $p(\hat{x}|x)$ because they cannot directly find the standard deviation of the noise probability distribution is not quite true. The real reason for using the conditional is the fact that estimating the partition function (z) does not have a tractable solution. This fact alone calls into question the soundness of the proposed method. Simply reducing the problem into estimating the standard deviation, completely ignores issue with the tractability of the partition function (z).

- The purported mathematical derivations are also quite unclear. The authors should clearly present the core theoretical problem with clear and concise formulations, prove the effectiveness of their proposed method, and draw clear theoretical conclusions. Most of the math shown in the paper is rather straightforward, without touching on the core issue for the use of conditional probabilities in the denoising process.

- Even if the theoretical background of the proposed work is sound, the proposed solution is rather incremental. The authors simply replace the encoder-decoder model of the DSM approach by two encoder-decoders and claim that it solves the problem. It is not clear what the original idea is, beyond simply estimating $\sigma$ rather than $\alpha$ in the original DSM method.

- There is also a lack of discussion on the significance of this approach, especially with respect to the contribution is the domain of diffusion models. Beyond very little improvements in accuracy of anomaly detection tasks, what is the major shortcoming of DSM that cannot be achieved by better training, and why this proposed method is needed. There are no empirical or experimental examinations to demonstrate the superiority of this approach over established diffusion models.

- The experimental design of the paper is also severely lacking in various aspects. Firstly, the experiments solely focus on anomaly detection in images. It is not clear if the performance shown in the experiments are directly related to the proposed score estimation or related to the anomaly detection method used. The paper also compares with three other diffusion models without clearly explaining how the other methods are utilized for the anomaly detection tasks. For example, did the authors replace the diffusion process in GLAD or ADSPR with their diffusion process, or are these results the direct end-to-end comparisons in anomaly detection. In addition, several of the reported AUROC values are 100% which hints at potential overfitting. There is no discussion on this issue.

**Questions:**

Please see weaknesses.

---

> ### Author Response · Authors · 2025-11-23
>
> The reviewer argues that our explanation of why diffusion models use conditional probabilities is inaccurate, claiming that the true reason lies in the intractability of the partition function Z. Based on this assumption, the reviewer questions the theoretical soundness of our method and suggests that we ignore the tractability issue of Z. Our method, however, is built upon the Gaussian assumption and does not involve the partition function Z. Relevant discussions regarding the Gaussian assumption can be found in the paper’s discussion section, and we have expanded this section by adding that “most diffusion models are indeed based on Gaussian distributions.” In practice, Gaussian-based formulations have consistently demonstrated strong empirical performance.
> Even under a Gaussian assumption, standard diffusion models still cannot obtain the explicit score. Under the Variance Preserving (VP) diffusion process, denoising score matching (DSM) can only estimate the conditional score rather than the marginal (explicit) score. As discussed in the paper, especially in the derivations around Eqs. (8)–(10), models trained with DSM and those trained with ESM are fundamentally different. For non-Gaussian cases, the discussion section also explains that one can adopt approaches similar to DIAD by projecting data into a Gaussian latent space. These considerations all indicate that DSM does not recover the explicit score even under an idealized Gaussian setting.
>
> For more precise wording, we have revised the statement in the manuscript to: “Under the Gaussian assumption, existing models are still limited to estimating the conditional score.”
> The core issue is that existing approaches can only rely on conditional probabilities, whereas our method enables estimation of the true marginal probability. Experiments show that this difference leads to a noticeable performance gap.
> The reviewer views the proposed method as incremental, suggesting that replacing one encoder–decoder with two does not constitute a meaningful contribution, and notes that the conceptual advance beyond estimating a different quantity from DSM is unclear. In fact, $\alphaα$ is merely a scalar parameter, whereas the variance $\sigmaσ$ is a matrix with each pixel assigned its own variance value. The experiments already demonstrate that our approach reaches state-of-the-art performance with significantly fewer diffusion steps, achieving substantial improvements in both accuracy and efficiency. Our method shows a clear improvement over DSM. Moreover, we observe differences between DSM and ESM in the generation process: Figure 3 illustrates this clearly—especially the upper bound of the variance—showing that stable variance greatly contributes to reliable image generation. Additionally, based on the theoretical foundation of this work, we further develop a variance-guided diffusion formulation. Empirically, DSM tends to produce unstable variance patterns that are difficult to remove in later steps, whereas models trained using ESM yield much more stable variance estimates.
>
> The reviewer questions whether DSM truly has limitations that cannot be resolved through better training, and argues that we do not provide sufficient empirical evidence demonstrating the superiority of our approach over established diffusion models. It is true that detection performance depends jointly on the detector and the generative model. Our primary goal is to show that the proposed diffusion model’s generative capability is strong enough to support state-of-the-art anomaly detection. Beyond anomaly detection, the model can also be applied to general image generation, medical imaging, and path planning, which we are currently developing. For the anomaly detection task specifically, our method provides explicit variance estimates, enabling the design of new detectors such as the $S_u$​ and $S_{KL}$ detectors described in first paragraph on page 6.
>
> The reviewer argues that our experiments focus only on image anomaly detection and do not isolate the effect of the proposed score estimation from that of the detection method. They also note that some baselines (e.g., GLAD, ADSPR) are not clearly described regarding how diffusion models are incorporated, and express concern that AUROC values of 100% may indicate overfitting. However, many existing approaches already achieve or nearly achieve 100% AUROC on certain benchmark categories, so such results cannot be taken as evidence of overfitting in our method. Our experiments strictly follow the evaluation protocols used by competing methods, and the appearance of perfect AUROC reflects the current state of the field rather than a flaw in our approach. In this work, anomaly detection serves only as a means to evaluate ESM. Extensions to broader application scenarios are underway, with related papers and patents to be submitted soon.

---

### Author Response · Authors · 2025-12-03

First, we thank the reviewers for their careful reading and constructive comments, which helped us substantially improve the manuscript. In response, we have made the following revisions and clarifications:
1.We redefined$​ \mathrm{ESM}_l$​ and $​ \mathrm{DSM}_l$​ ​ (Eqs. (6)–(7)) to make the distinction between ESM and DSM explicit. Conceptually ESM and DSM are not the same — they coincide only when$​l=2$​  (the result in Vincent (2011) is limited to the $​ l=2$​  case). This clarification does not change our original conclusions, although it explains why certain choices can lead to degraded performance.
2.We supplemented the discussion of variance-based noise injection (Eq. (11)). We show that adding noise guided by variance does not violate the SDE formulation; in fact, this variance-guided noise satisfies Eq. (12). The full proof is provided in the appendix.
3.On the Gaussianity issue: many recent works already adopt Gaussian assumptions in a latent space (e.g., Mao et al. (2020)), and this can also be achieved via an autoencoder-style projection as in DiAD (He et al. (2024)). We have updated the discussion accordingly and additionally clarified that diffusion-generation procedures (e.g., DDPM) start from Gaussian noise and can produce images that are effectively modeled as Gaussian in the generative process, regardless of whether the raw input images strictly follow a Gaussian distribution.
4.Regarding the experiments: because our method uses multiple detection heads whose coefficients are hyperparameters, we tune those coefficients using defect samples from the training set only (no test-set involvement), so there is no test data leakage. We emphasize that the experiment is intended to demonstrate the applicability of the proposed method to defect detection as one downstream application; the method itself is more generally applicable to other downstream tasks.
We believe these revisions substantially address the reviewers’ concerns. Empirically, the proposed method demonstrates properties not present in prior approaches — for example, a stable variance range (see Fig. 3 and the discussion in Sec. 4.2) — which may enable further improvements in diffusion-model performance.

We would also like to politely note a few possible misunderstandings raised by some reviewers:
(1)Reviewer ewrt: concerning the case of very small variance, our loss formulation prevents the variance from collapsing to zero.
(2)Reviewer ewrt: regarding mixture-of-Gaussians — our method does not require all images to follow a single Gaussian. For instance, two inputs $X_1$​ and $X_2$may each follow $\mathcal{N}(\mu_1,\sigma_1)$ and $\mathcal{N}(\mu_2,\sigma_2)$ respectively; this can be interpreted as a (discrete) mixture of Gaussians with appropriate weights.
(3)Reviewer QycL06: concerning the claim that the difference with DSM is minor — in DSM the scalar $\alphaα$ is a single parameter, while our method estimates a variance $\sigmaσ$ as a matrix (i.e., a per-pixel variance). The per-pixel variance clearly carries substantially more information than a single scalar parameter.
We hope this summary helps the AC and reviewers — full details and our formal responses can be found in the revised manuscript and accompanying response file.

---

### Meta-Review · Area_Chair_FQ6Y · 2025-12-09

**Summary:**

The paper has serious theoretical flaws, mainly manifested as: 1) The meaning of the ESM formula (eq 3) is incorrect; 2) There are issues with the calculation of formulas (6) and (7); 3) Important process derivation is missing between eq (9) and eq(10).
The experiments in the paper are insufficient to demonstrate the effectiveness of the method。

**Reviewer Concerns:**

Reviewer QycL and EWrT questioned the theoretical flaws in the paper, but the author did not provide effective responses. Additionally, for the question of reviewer Z6vw and RhCF， the author did not conduct any experimental supplements.

**Reviewer Scores:**

Reviewer QycL and EWrT will not increase the score because the reviewers' questions have not been addressed;
Reviewers Z6vw and RhCF will decrease the score because the authors have not provided experimental supplements to the reviewers' questions.

---

### Decision · Program_Chairs · 2026-01-26

Reject